# Novel Digital Measurement Technique to Analyze the Palatine Suture Expansion after Palatine Rapid Maxillary Expansion Technique

**DOI:** 10.3390/jpm11100962

**Published:** 2021-09-27

**Authors:** Mariano Requena Pérez, Álvaro Zubizarreta-Macho, Pedro Colino Gallardo, Alberto Albaladejo Martínez, Daniele Garcovich, Alfonso Alvarado-Lorenzo

**Affiliations:** 1Department of Surgery, Faculty of Medicine and Dentistry, University of Salamanca, 37008 Salamanca, Spain; idu018871@usal.es; 2Department of Orthodontics, Faculty of Medicine and Dentistry, University of Salamanca, 37008 Salamanca, Spain; albertoalbaladejo@usal.es (A.A.M.); kuki@usal.es (A.A.-L.); 3Department of Implant Surgery, Faculty of Health Sciences, Alfonso X el Sabio University, 28691 Madrid, Spain; 4Department of Orthodontics, Faculty of Faculty of Health Sciences, European University Miguel de Cervantes, 47012 Valladolid, Spain; pgo.pcolino@odontologia.ucam.com; 5Department of Dentistry, European University of Valencia, Passeig de lÁlbereda 7, 46010 Valencia, Spain; daniele.garcovich@universidadeuropea.es

**Keywords:** orthodontics, palatine suture expansion, rapid maxillary expansion, digital dentistry, McNamara appliance

## Abstract

The aim of the study was to validate a novel digital measurement method to quantify the volume of the midpalatal suture after rapid maxillary expansion (RME). Material and methods: Eight patients with maxillary skeletal transverse deficiency were submitted to palatine suture expansion using the McNamara orthodontic appliance during a period of nine months. After 30 days of treatment, all patients were exposed postoperatively to a cone-beam computed tomography (CBCT) scan. Afterwards, the scans were uploaded into the three-dimensional orthodontic-planning software to allow the volumetric assessment of the palatine suture expansion through palatine rapid maxillary expansion using a McNamara appliance. The repeatability was analyzed by repeating twice the measures by a single operator and reproducibility was analyzed by repeating three times the measures by two operators, and Gage R&R statistical analysis was performed. Results: The expansion of the midpalatal suture by means of the rapid maxillary expansion technique, in terms of digital volume measurement, showed a repeatability value of 0.09% and between the two operators a reproducibility value of 0.00% was shown. Conclusions: The novel measurement technique demonstrated a high repeatability and reproducibility rate for volume assessment of the palatine suture expansion through palatine rapid maxillary expansion technique.

## 1. Introduction

Palatine suture expansion through rapid maxillary expansion (RME) procedures has been widely spread since Angell described the technique in 1860 to correct the maxillary compression in growing patients [1,2,3]. Subsequently, this therapeutic procedure has experienced a continuous development and it has been recommended for posterior crossbite, tooth crowding [4], respiratory distress and class II malocclusion due to posterior crossbite generated by mandibular advancement [5]. In addition, RME procedures allow the expansion of the middle palatal suture by expanding the palatal bones and consequently increasing the maxillary width. Therefore, RME procedures have evidenced to correct both dental and maxillary transverse discrepancies, even the maxillary constriction related to respiratory disorders such as obstructive sleep apnea (OSA) in growing patients [5,6,7,8,9]. Previous studies have analyzed the dento-skeletal [10] and morphological outcomes [11] of RME procedures using a computed tomography (CT) scan [12] or cone-beam computed tomography (CBCT) scan [4], and highlighted both width and depth increasing of the maxilla and also an increase in the inclination of the dentoalveolar process after RME procedures [1,13,14,15,16,17]. In addition, the maxillary expansion after RME procedures has also been quantified by linear [18] and angular measurements [19] and even by analyzing the area of the palate surface [20,21,22,23]. However, Marini et al. reported that the palatal volume increases in patients treated with RME procedures and recommended the application of three-dimensional (3D) measurement techniques to evaluate the skeletal changes [2]. In addition, Sam et al. showed that the cephalometric analysis based on 3D measurement techniques was more reliable than conventional two-dimensional (2D) measurement techniques [24]. Moreover, Angelieri et al. suggested a procedure to individually diagnose the most predictable palatine expansion approach, regarding the bone healing stage of the midpalatal suture assessed by CBCT scan, concluding that initial stages (A and B) are associated to bone healing in progress, intermediate stages (C and D) are associated to a partially closed suture and advanced stage (E) requires a surgical approach [25].

The aim of this work was to validate a measurement method to quantify the volume of the midpalatal suture after RME using the McNamara orthodontic appliance. The null hypothesis (H0) of the present study was as follows: The novel digital measurement technique for quantifying the volumetric expansion of the palatine suture is neither repeatable nor reproducible.

## 2. Materials and Methods

### 2.1. Study Design

An experimental study was conducted at the University of Salamanca (Salamanca, Spain) and Alfonso X el Sabio University (Madrid, Spain) between November 2020 and April 2021, and was authorized by the Ethical Committee of the Faculty of Dentistry, Alfonso X el Sabio University (Madrid, Spain) in November 2020 (process no. 09/2020). Patients were informed of the ethical procedures of the study and gave informed consent for the processing of CBCT scans.

### 2.2. Clinical Procedure

Ten patients (4 male and 6 female) were submitted to palatine suture expansion by RME procedures through McNamara appliance, which is a tooth-supported orthodontic appliance that consists of two acrylic parts attached in the posterior tooth and connected by an expansion screw in the midpalatal region (Figure 1). The inclusion criteria were patients between 9–17 years of age and in good health, who presented signs and symptoms of maxillary skeletal transverse deficiency and A or B stage midpalatal suture maturation according to Angilieri’s classification [24]. Patients who were not within the selected age range, presented a different bone maturation stage (C, D or E), presented systemic conditions, craniofacial syndromes, malformations or had a different diagnosis or treatment plan were excluded. Afterwards, the rapid maxillary expansion procedure through McNamara was made by taking oral impressions with two-phase silicone (Aquasil, Dentsply, Erlangen, Germany). The expansion device was 13 mm long with 1.5 mm thick arms and 0.8 mm full turn (Leone, Florence, Italy) and glass-ionomer cement was used to adhere on the upper molars and premolars. The expansion protocol (Figure 1) was 1/4 turn per day for 15 days (0.4 mm per day), which was the active period of disjunction [26]. The appliance was subsequently kept for up to 9 months of passive retention to avoid relapses.

### 2.3. Measurement Procedure

After the palatine suture expansion was achieved by rapid maxillary expansion appliances through disjunctor, all the patients were submitted to a postoperative CBCT scan (WhiteFox, Satelec, Merignac, France) with the following exposure parameters: 105.0 kV peak, 8.0 mA, 7.20 s, and a field of view of 15 mm × 13 mm. The position of the patient’s head was standardized so that the Frankfort plane was parallel, and the midsagittal plane was perpendicular to the ground. Afterwards, datasets from the postoperative CBCT scans (WhiteFox, Satelec, Merignac, France) were uploaded into the three-dimensional orthodontic-planning software (NemoFAB 3D NemoStudio^®^ 19.2.0 uv 89 2KT, Nemotec; Madrid, Spain) to perform the volumetric measurement of the palatine suture expansion. Subsequently, the midpalatal suture was aligned to the sagittal axis and the palatal plane was aligned to the axial axis (Figure 2).

Then, the geometric prism of the expanded palatine suture was selected and isolated by analyzing the boundaries of the palatine suture expansion in transversal (Figure 3A), sagittal (Figure 3B), coronal (Figure 3C) plane and the three-dimensional reconstruction of the CBCT scan (WhiteFox, Satelec, Merignac, France) (Figure 3D). Subsequently, the anatomical references that delimited the boundaries of the geometric prism after the palatine suture expansion were identified by the two anterior nasal spines and the two posterior nasal spines.

Afterwards, a reference point (seed point) was placed inside the selected and isolated geometric prism of the expanded palatine suture at the transversal (Figure 4A), sagittal (Figure 4B) and coronal (Figure 4C) plane and the three-dimensional reconstruction (Figure 4D). This was to ensure a tissue density with a tolerance range of 500 Hounsfield units (HU), corresponding to the soft tissue density formed after the palatine suture expansion was performed according to the study of Buzatu et al. who measured the density of the palatal suture after RME procedures ranging between 128.56 HU (middle segment in women) and 398.07 HU (posterior segment in men) [27]. In the present study, a 500 HU tolerance range was established to ensure that the entire palatal suture expansion was registered inside the previously defined geometric prism.

Finally, the volumetric measurement (mm^3^) of the palatine suture expansion after RME procedures through McNamara appliance was performed using the three-dimensional orthodontic-planning software (NemoFAB 3D NemoStudio^®^ 19.2.0 uv 89 2KT, Nemotec; Madrid, Spain).

In addition, lineal measurements of the palatine suture expansion were also performed in order to analyze and compare the repeatability and reproducibility of the previously described conventional measurement technique [28,29] with the novel digital measurement method to quantify the volume of the midpalatal suture after rapid RME procedures. After uploading the images into three-dimensional orthodontic-planning software (NemoFAB 3D NemoStudio^®^ 19.2.0 uv 89 2KT, Nemotec; Madrid, Spain) (Figure 5A), the midpalatal suture was aligned to the sagittal axis (Figure 5B) and the palatal plane was aligned to the axial axis (Figure 5C). Subsequently, horizontal lineal measurements from the cortex of the disjunction were performed at the central incisors, canines and first permanent molars (Figure 5D).

### 2.4. Validation of the Repeatability and Reproducibility

In order to validate the repeatability and reproducibility of the digital measurement technique and lineal measurement technique to quantify the palatine suture expansion after RME procedures through McNamara appliance, the measures were repeated three times by two operators (Operator A and Operator B) in each one of the eight patients and Gage R&R statistical analysis was performed.

### 2.5. Statistical Analysis

Statistical analysis of the measurement variables was conducted using SAS 9.4 (SAS Institute Inc., Cary, NC, USA) and R (R Foundation for Statistical Computing, Vienna, Austria) and descriptively expressed as means and standard deviations (SD). The repeatability and reproducibility of the digital measurement method and lineal measurement technique were analyzed by Gage R&R statistical analysis. The statistical significance was set at *p* < 0.05.

## 3. Results

The means and SD values for the volume of the palatine suture expansion through palatine rapid maxillary expansion technique between operators are displayed in Table 1 and Figure 6 and Figure 7

The Gage R&R statistical analysis of the digital measurement technique in terms of the volume of the palatine suture expansion through palatine rapid maxillary expansion technique showed a repeatability value of 0.09%. Repeatability values below 1% are necessary to demonstrate high repeatability; therefore, the morphometric measurement technique demonstrated a high repeatability rate for area measurement (Figure 6 and Figure 7). In addition, the Gage R&R statistical analysis of the digital measurement technique in terms of the volume of the palatine suture expansion through palatine rapid maxillary expansion technique performed by the two operators showed a reproducibility value of 0.00%. Reproducibility values below 1% are necessary to demonstrate high reproducibility; therefore, the morphometric measurement technique demonstrated a high reproducibility rate for volume of the palatine suture expansion through palatine rapid maxillary expansion technique (Figure 6 and Figure 7).

The means and SD values for the lineal distance of the palatine suture expansion through palatine rapid maxillary expansion technique between operators are displayed in Table 2 and Figure 8 and Figure 9.

The Gage R&R statistical analysis of the lineal measurement technique in terms of the distance of the palatine suture expansion through palatine rapid maxillary expansion technique showed a repeatability value of 73.68% and repeatability values below 1% are necessary to demonstrate high repeatability rate (Figure 8 and Figure 9). In addition, the Gage R&R statistical analysis of the lineal measurement technique in terms of the distance of the palatine suture expansion through palatine rapid maxillary expansion technique performed by the two operators showed a reproducibility value of 0.00%. Reproducibility values below 1% are necessary to demonstrate high reproducibility; therefore, the lineal measurement technique demonstrated a high reproducibility rate for the lineal measurement of the palatine suture expansion through palatine rapid maxillary expansion technique (Figure 8 and Figure 9).

In summary, the novel digital technique is a reproducible and repeatable measurement method for quantifying the volumetric expansion of the palatine suture after rapid maxillary expansion technique, but the lineal measurement technique is not a repeatable but reproducible measurement method.

## 4. Discussion

The results obtained in the present study rejected the null hypothesis (H_0_) that states that the novel digital measurement technique for quantifying the volumetric expansion of the palatine suture is neither repeatable nor reproducible.

The anatomical references commonly used in orthodontics for 2D measurement methods require skill and precision by the operator to obtain reliable results [30]. Moreover, Lagravere et al. conclude that 3D measurement methods using CBCT scan are more reliable and reproducible than 2D measurement methods and that the variability between cephalometric anatomical references can be greater than 1 mm [31]. However, van Vlijmen et al. reported that some anatomical references plotted for 2D measurement methods are not applicable in 3D measurement methods [32]. Therefore, the CBCT scan should be used not only to verify the anatomical references plotted in 2D radiographies but also to create new volumetric measurement methods.

Numerous authors have already demonstrated the efficacy of RME in the transverse gain of the maxilla through different pre and postoperative measurement processes using sagittal, coronal and axial planes given by CBCT scan so that numerous measures can be obtained through linear records, angulations between planes and surface areas [19,20,21,33,34,35]. However, despite the use of CBCT technology that allows for the three-dimensional analysis of cranial structures, to our knowledge there are no studies that measure the maxillary volume obtained after RME procedures.

Christie et al. analyzed and quantified the effects following RME procedures at both the skeletal and dentoalveolar levels to discern to what degree the expansion corresponds to skeletal structures through opening of the midpalatal suture or whether it is due to tilting of the dentoalveolar processes [36]. In addition, they reported a mean increase of 40.65% (3.33 mm), 44.08% (3.49 mm), 46.73% (3.83 mm) and 46.83% (3.62 mm) of the maxillary basal bone and an opening of the midpalatal suture of 52.82% (4.33 mm), 53.23% (4.36 mm), 54.35% (4.46 mm) and 52.77% (4.33 mm) at the first permanent molars, second deciduous molars, first deciduous molars and deciduous canines, respectively. Lione et al. reported a transverse skeletal augmentation between 25.6% and 57.5%, depending on the device used, and the dental expansion was between 77% and 42% [16]. Christie et al. also analyzed the dentoalveolar effects indicating that the right first upper molar tilted buccally by an average of 6.2° and the left first upper molar tilted buccally by an average of 5.6° [32]. Lo Giudice shows an increase in coronal tilt of between 0.98° and 4.03° for tooth-supported expanders compared to 0.24°–0.78° for bone-supported devices [10]. In addition, periodontal manifestations such as an increase in cortical bone of 0.6 mm on each side may occur in denture-supported dentition [15]. However, Liu et al. highlighted 2D measurement methods to assess the midpalatal suture after RME procedures, although only half of the selected reports showed data for the anterior and posterior areas. The anterior expansion of the palatine suture varies between 2.42 mm and 4 mm (34.6%–50% of the total expansion of the screw) and between 0.84 mm and 2.88 mm (12–36% of the total expansion of the screw) at the posterior region. Five studies measured changes at the level of the canines and the first permanent molars ranging between 1.52 mm and 4.3 mm [29]. Cantarella suggests the possibility of differentiating the measurements of the anterior and posterior nasal spine and hence evaluate the effects on the midpalatal suture by quantifying the parallelism of the latter. The measurement of the opening in the posterior nasal spine (4.3 mm) was 90% of the opening in the anterior nasal spine (4.8 mm) [14]. In addition, Garret et al. reported that the skeletal expansion of the maxilla had a triangular pattern with a wider base in the anterior region, representing 55% of the total expansion at the level of the first upper premolars, 45% at the second upper premolars and 38% at the first upper molars; however, they reported an alveolar inclination of 6% at the first upper premolars, 9% at the second upper premolars and 13% at the first upper molars [1]. However, Mosleh et al. evidenced no statistical significance for the bucco-palatal tilt of first upper permanent molars [37]. All these data highlight the great controversy that exists regarding the skeletal and dentoalveolar effects that occur after RME procedures; therefore, it is mandatory to create a repeatability and reproducibility measurement procedure to unify the methodological design of the research. Therefore, Koçer et al. conclude that volumetric measurements can be applied and combined with linear measurements to obtain a direct clinical application in the diagnosis of oedema caused by RME procedures to improve the treatment outcome and prognosis [38].

Traditionally, the nasal width of the maxillary bone has been measured in frontal (anteroposterior) cephalometric analysis using the jugale points, the distance between them being used to estimate the transverse dimension of the maxilla [39]. However, due to the superimposition of numerous structures, some points can be difficult to locate. Many authors have tried to avoid this by using axial or coronal slices [10,14,19,32,33]. This leads to the plotting of a larger number of points and linear or angular measurements [10,16,40]. The limitations of the study were to perform a larger number of measurements with different types of patients, separating them by age and gender. Moreover, to perform the measurements with different groups of operators to see if experience influences the measurement and use different types of computer programmes to see what influence they have on the measurements of the volume of the midpalatal suture.

## 5. Conclusions

In conclusion, the results show that the novel digital measurement technique is reproducible and repeatable and is valid for quantifying the volume of the palatine suture expansion through palatine rapid maxillary expansion technique.

## Figures and Tables

**Figure 1 jpm-11-00962-f001:**
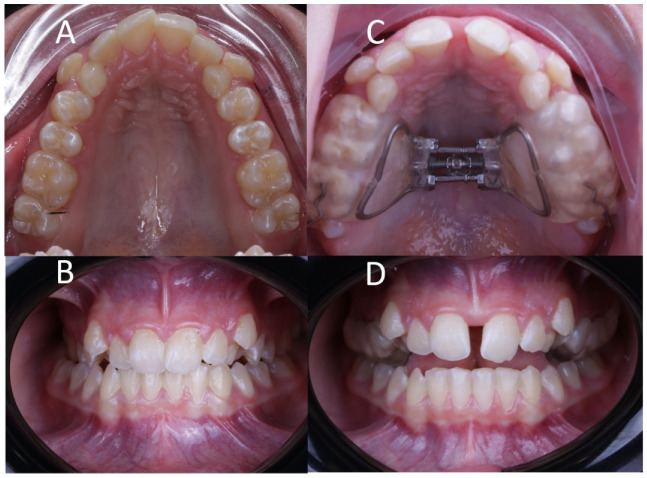
(**A**) Preoperative occlusal and (**B**) frontal view, and (**C**) postoperative occlusal and (**D**) frontal view after RME treatment with McNamara appliance.

**Figure 2 jpm-11-00962-f002:**
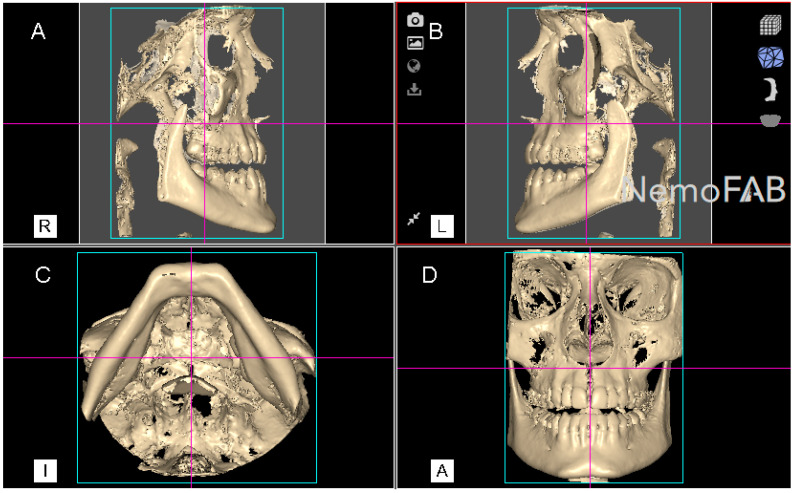
(**A**) Right lateral, (**B**) left lateral, (**C**) bottom and (**D**) frontal view of the postoperative cone-beam computed tomography (CBCT) scans, aligned to the Frankfort plane and the midsagittal plane was perpendicular to the ground.

**Figure 3 jpm-11-00962-f003:**
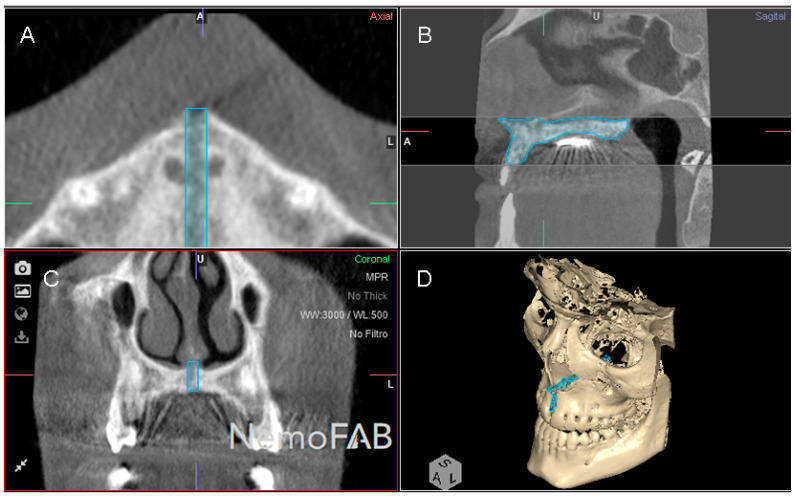
(**A**) Boundaries of the palatine suture (blue line) after the RME procedure through McNamara appliance was performed, at transversal plane, (**B**) sagittal plane, (**C**) coronal plane and (**D**) at three-dimensional reconstruction of the CBCT scan.

**Figure 4 jpm-11-00962-f004:**
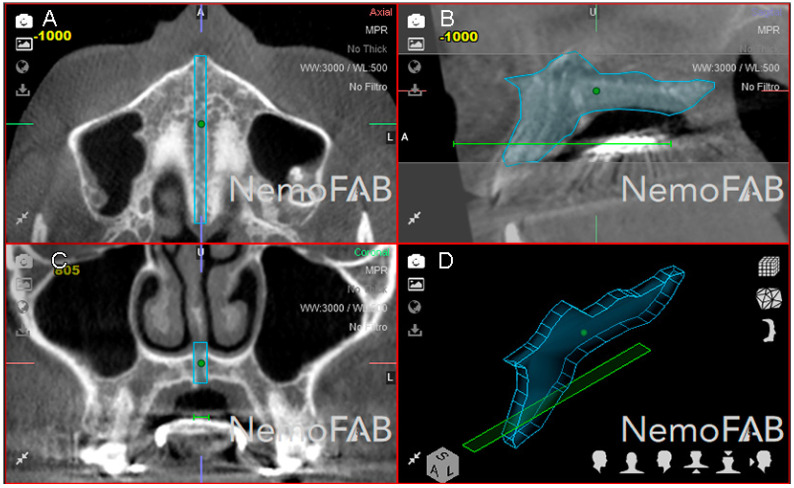
(**A**) Boundaries of the palatine suture (blue line) after the RME procedure through McNamara appliance with the seed point inside, at transversal plane, (**B**) sagittal plane, (**C**) coronal plane and (**D**) at three-dimensional reconstruction.

**Figure 5 jpm-11-00962-f005:**
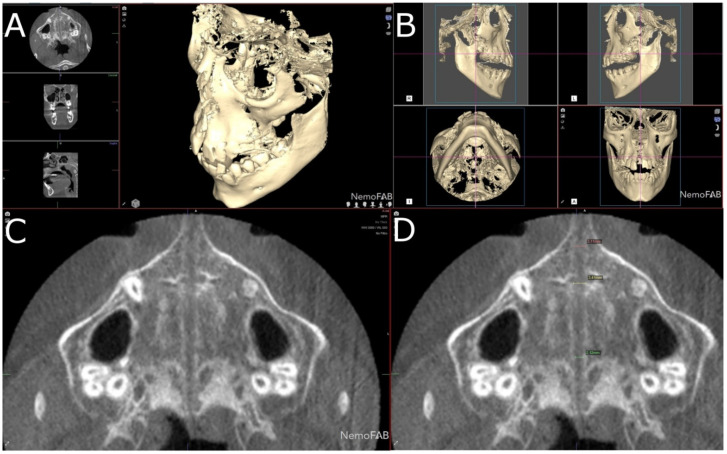
(**A**) CBCT scan uploaded to the three-dimensional orthodontic-planning software, (**B**) alignment procedure of the CBCT scan, (**C**) alignment and (**D**) measurement procedure of the palatine suture expansion at the central incisors, canines and first permanent molars.

**Figure 6 jpm-11-00962-f006:**
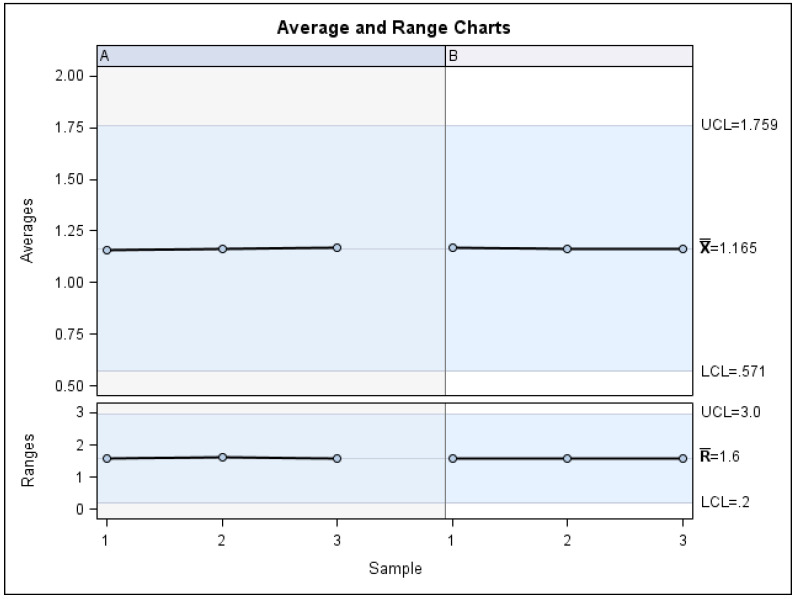
Charts of the average of the three measurements of the volumetric measurement (mm^3^) of the palatine suture expansion after RME procedures through McNamara appliance. Left column (A) is referred to Operator 1 and right column (B) is referred to Operator 2.

**Figure 7 jpm-11-00962-f007:**
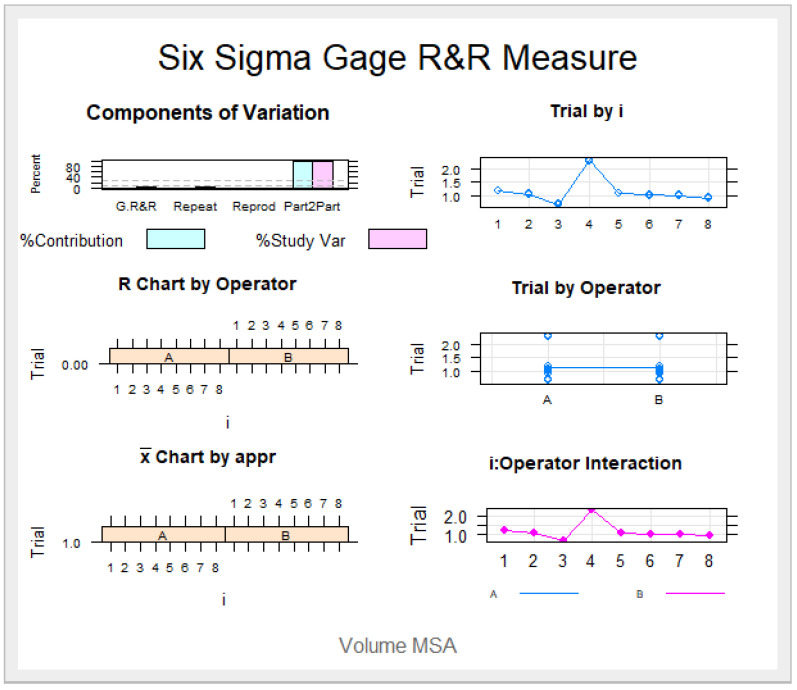
The graph shows the relationship of the volume measurement analysis after palatal suture expansion following RME procedures through the McNamara appliance, with the contribution graph of each component to the total variance (Variance Components), mean control chart and range control chart (R Chart by Operator and x Chart by appr), each measurement point on the graph (Assay by I and Assay by Operator) and the interaction between the operators (i:Operator interaction).

**Figure 8 jpm-11-00962-f008:**
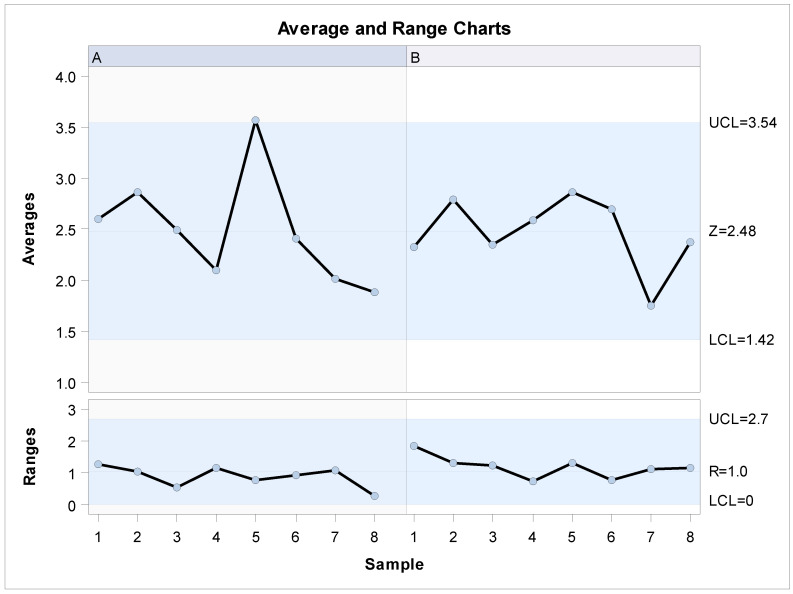
Charts of the average of the three measurements of the lineal measurement (mm) of the palatine suture expansion after RME procedures through McNamara appliance. Left column (A) is referred to Operator 1 and right column (B) is referred to Operator 2.

**Figure 9 jpm-11-00962-f009:**
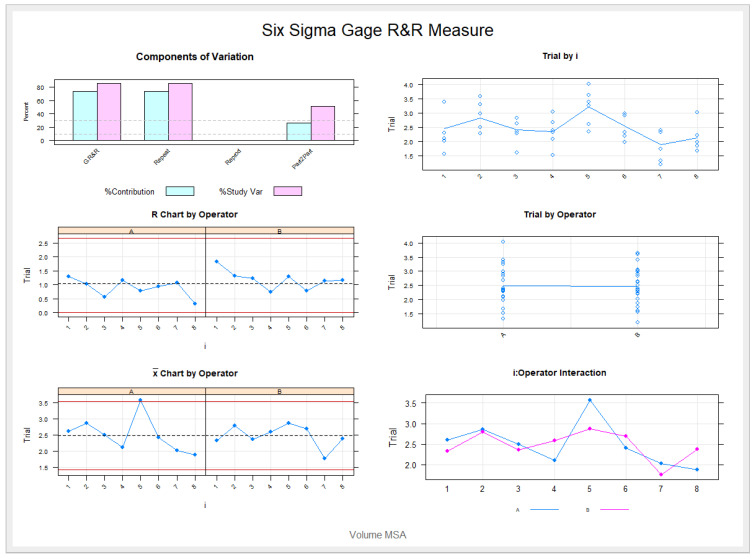
The graph shows the relationship of the lineal measurement analysis after palatal suture expansion following RME procedures through the McNamara appliance, with the contribution graph of each component to the total variance (Variance Components), mean control chart and range control chart (R Chart by Operator and x Chart by appr), each measurement point on the graph (Assay by I and Assay by Operator) and the interaction between the operators (i:Operator interaction). Operator 1 is defined as letter A and Operator 2 is defined as letter B.

**Table 1 jpm-11-00962-t001:** Descriptive statistics of the area of the volume of the palatine suture expansion through palatine rapid maxillary expansion technique between operators.

Operator	I	n	Mean	SD	Minimum	Maximum
A	1	3	1.190	0.010	1.180	1.200
	2	3	1.060	0.010	1.050	1.070
	3	3	0.693	0.015	0.680	0.710
	4	3	2.290	0.017	2.280	2.310
	5	3	1.107	0.015	1.090	1.120
	6	3	1.023	0.006	1.020	1.030
	7	3	1.007	0.021	0.990	1.030
	8	3	0.940	0.000	0.940	0.940
B	1	3	1.193	0.012	1.180	1.200
	2	3	1.063	0.015	1.050	1.080
	3	3	0.697	0.015	0.680	0.710
	4	3	2.290	0.026	2.260	2.310
	5	3	1.103	0.015	1.090	1.120
	6	3	1.037	0.006	1.030	1.040
	7	3	1.017	0.015	1.000	1.030
	8	3	0.927	0.021	0.910	0.950

I: Operator interaction; SD: standard deviation.

**Table 2 jpm-11-00962-t002:** Descriptive statistics of the lineal distance of the palatine suture expansion through palatine rapid maxillary expansion technique between operators.

Operator	I	n	Mean	SD	Minimum	Maximum
A	1	3	2.603	0.696	2.110	3.400
	2	3	2.863	0.525	2.290	3.320
	3	3	2.490	0.305	2.280	2.840
	4	3	2.103	0.585	1.520	2.690
	5	3	3.570	0.414	3.260	4.040
	6	3	2.410	0.469	1.980	2.910
	7	3	2.020	0.599	1.330	2.400
	8	3	1.880	0.173	1.680	1.980
B	1	3	2.330	0.956	1.560	3.400
	2	3	2.793	0.707	2.280	3.600
	3	3	2.353	0.660	1.600	2.830
	4	3	2.590	0.400	2.320	3.050
	5	3	2.870	0.688	2.350	3.650
	6	3	2.697	0.432	2.200	2.980
	7	3	1.753	0.565	1.200	2.330
	8	3	2.373	0.598	1.860	3.030

I: Operator interaction; SD: standard deviation.

## Data Availability

Data available on request due to restrictions eg privacy or ethical.

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
