# Peer review of "Novel Digital Measurement Technique to Analyze the Palatine Suture Expansion after Palatine Rapid Maxillary Expansion Technique"

_jpm, 2021, doi:10.3390/jpm11100962_

Round 1

Reviewer 1 Report

Dear authors, 

My main comments after reading your study are:

  1. I think that you have to further explain what the is the benefit of using a volumetric versus a linear measurement for the assessment of palatal expansion. It is an innovative technique, but how does it add to current knowledge? This is not clear in your manuscript.
  2. The height of the palatal bone differs between individuals, so comparing volumetric changes between subjects would probably not be appropriate to compare different individuals in terms of treatment efficacy. What is/are the possible applications of this technique in general? Please clarify.
  3. Please provide a description of the McNamara appliance in your methods section. You need to briefly describe the different parts of the  appliance as well as whether it is teeth borne or bone borne, so as that the reader can understand the way it works.
  4. Was a power/sample size calculation included in your project?
  5. How did you define 500 HUs as the representative value for soft tissue in these scans?

Thank you and I look forward to your reply.

Author Response

Dear Reviewer 1,

I’m pleased to resubmit the manuscript of the work entitled, “Novel Digital Measurement Technique to Analyze the Palatine Suture Expansion After Palatine Rapid Maxillary Expansion Technique”

Reviewer 1: Moderate English changes required

Response: In order to adapt to the reviewer's 1 comments, we have send the manuscript to a specialized traductor.

Reviewer 1: I think that you have to further explain what the is the benefit of using a volumetric versus a linear measurement for the assessment of palatal expansion. It is an innovative technique, but how does it add to current knowledge? This is not clear in your manuscript.

Response: In order to adapt to the reviewer's 1 comments, we clarified in the Discussion section that the anatomical references commonly used in orthodontics for 2D measurement methods require skill and precision by the operator to obtain reliable results. In addition, the results derived from the present study showed that the lineal measurement method results less accurate and it is not repeatable and reproducible when comparing with the novel digital measurement technique.

Reviewer 1: The height of the palatal bone differs between individuals, so comparing volumetric changes between subjects would probably not be appropriate to compare different individuals in terms of treatment efficacy. What is/are the possible applications of this technique in general? Please clarify.

Response: In order to adapt to the reviewer's 1 comments, we clarified that he height and width of the jaws are different in each patient; therefore, the application of this technique is not only to see the volume generated by breaking the palatal suture but also to analyze all kind of patients conditions by means of a repeatable and reproducible measurement technique and compare different types of appliance such as osteosoporated which according to the literature generate a greater opening of the palatal suture in comparison with tooth-supported devices such as the Mcnamara. We can also compare the volume of the mid palatal suture with a larger airway volume and improved patient breathing.

Reviewer 1: Please provide a description of the McNamara appliance in your methods section. You need to briefly describe the different parts of the  appliance as well as whether it is teeth borne or bone borne, so as that the reader can understand the way it works.

Response: In order to adapt to the reviewer's 1 comments, we have described the McNamara appliance in the Material and Methods section.

Reviewer 1: Was a power/sample size calculation included in your project?

Response: In order to adapt to the reviewer's 1 comments, we clarify that for Gage R&R statistical analysis it is not usual to carry out these calculations, and there is no standard. We have published previous articles with this statistical analysis using 6 measurements by operator (Zubizarreta-Macho Á, Triduo M, Alonso Pérez-Barquero J, Guinot Barona C, Albaladejo Martínez A. Novel Digital Technique to Quantify the Area and Volume of Cement Remaining and Enamel Removed after Fixed Multibracket Appliance Therapy Debonding: An In Vitro Study. J Clin Med. 2020 Apr 12;9(4):1098) and even 3 measurements by operator (Belanche-Monterde A, Albaladejo Martínez A, Alvarado Lorenzo A, Curto A, Alonso Pérez Barquero J, Guinot-Barona C, Zubizarreta-Macho A. A repeatable and reproducible digital method to quantify the cement excess and enamel loss after debonding lingual multibracket appliance therapy. Appl. Sci. 2021, 11, 1295. doi.org/10.3390/app11031295.).

Reviewer 1: How did you define 500 HUs as the representative value for soft tissue in these scans?

Response: In order to adapt to the reviewer's 1 comments, we have clarified the tolerance range selection in the Material and Methods section.

We take this opportunity to thank the recommendations and suggestions made by the reviewers to improve the document.

Yours sincerely,

Reviewer 2 Report

According to the authors there are no studies that  measure the maxillary volume obtained after rapid maxillary expansion procedures. In general, the manuscript is well written, although there are some issues that  would be interesting to clarify.

In the abstract authors state that the measurements were repeated twice but in M&M they say that “measures were repeated three times by two operators “ Please check this situation.

The null hypothesis is confusing .

How was the sample size calculated?

What was the version of the software used?

Its not clear how the measurement area is delimited. A precise description of the procedure would be important to allow its replication.

Why didn't you perform a previous CBCT and compared the volumetric changes?

In addition to repeatability and reproducibility don´t you think it would be important to compare this method with others?

What is the usefulness/clinical relevance of the presented method?

Instead of suture palatine  you should say  palatine suture.

At the end of the manuscript also check this situation:

Institutional Review Board Statement: Not applicable.

Informed Consent Statement: Not applicable.

 that is not in accordance with the description at the beginning of M&M. In addition, as patients are under 18, consent must also be given by their parents or legal guardians.

Author Response

Dear Reviewer 2,

I’m pleased to resubmit the manuscript of the work entitled, “Novel Digital Measurement Technique to Analyze the Palatine Suture Expansion After Palatine Rapid Maxillary Expansion Technique”

Reviewer 2: English language and style are fine/minor spell check required

Response: In order to adapt to the reviewer's 2 comments, we have send the manuscript to a specialized traductor.

Reviewer 2: In the abstract authors state that the measurements were repeated twice but in M&M they say that “measures were repeated three times by two operators “ Please check this situation.

Response: In order to adapt to the reviewer's 2 comments, we have corrected the Abstract section.

Reviewer 2: The null hypothesis is confusing .

Response: In order to adapt to the reviewer's 2 comments, we have clarified the null hypothesis.

Reviewer 2: How was the sample size calculated?

Response: In order to adapt to the reviewer's 2 comments, we clarify that that for this type of statistical analysis the recommended value is 10 samples with 2 replicates or 8 samples with 3 replicates.

Reviewer 2: What was the version of the software used?

Response: In order to adapt to the reviewer's 2 comments, we have clarified the version of the software.

Reviewer 2: Its not clear how the measurement area is delimited. A precise description of the procedure would be important to allow its replication.

Response: In order to adapt to the reviewer's 2 comments, we have described the anatomical points that delimited the boundaries of the geometric prism after the palatine suture expansion.

Reviewer 2: Why didn't you perform a previous CBCT and compared the volumetric changes?

Response: In order to adapt to the reviewer's 2 comments, we clarify that the suture palatine of all patients is fully closed before the RME procedure; therefore, the value is “0” to all patients and comparison would be not possible. However, we have improved the article by comparing the novel digital measurement technique with the previously used lineal measurement technique in order to analyze the repeatability and reproducibility of both techniques.

Reviewer 2: In addition to repeatability and reproducibility don´t you think it would be important to compare this method with others?

Response: In order to adapt to the reviewer's 2 comments, we have improved the article by comparing the novel digital measurement technique with the previously used lineal measurement technique in order to analyze the repeatability and reproducibility of both techniques.

Reviewer 2: What is the usefulness/clinical relevance of the presented method?

Response: In order to adapt to the reviewer's 2 comments, we have we have clarified in the Discussion section that the anatomical references commonly used in orthodontics for 2D measurement methods require skill and precision by the operator to obtain reliable results. In addition, the results derived from the present study showed that the lineal measurement method results less accurate and it is not repeatable and reproducible when comparing with the novel digital measurement technique. In addition, this measurement method could be apply to analyze the palatine suture expansion after different RME procedures.

Reviewer 2: Instead of suture palatine  you should say  palatine suture.

Response: In order to adapt to the reviewer's 2 comments, we have changed the words.

Reviewer 2: At the end of the manuscript also check this situation: Institutional Review Board Statement: Not applicable. Informed Consent Statement: Not applicable.  that is not in accordance with the description at the beginning of M&M. In addition, as patients are under 18, consent must also be given by their parents or legal guardians.

Response: In order to adapt to the reviewer's 2 comments, we have corrected the sentences.

We take this opportunity to thank the recommendations and suggestions made by the reviewers to improve the document.

Yours sincerely,

Reviewer 3 Report

The authors aim to find the repeatability and reproducibility of a novel measurement technique. I feel major improvements are required in the following areas:

  1. There is no control group.
  2. The null hypothesis does not correspond to the aim of the study.
  3. Both in the introduction and discussion sections, the authors talked about RME. However, the focus of the study is the novel technique.
  4. The limitations mentioned are actually limitations of the technique, not of the study.

Suggestions to the authors:

  1. The study will make more sense if you can compare the novel technique with a standard technique.
  2. Please be clearer about the null hypothesis. Your hypothesis should answer your primary research question.
  3. Please focus your intro and discussion on the novel technique you are presenting and how is it different from the standard techniques. RME is not the focus here.
  4. Please correctly identify the limitations of the study design. Limitations of the technique are not equivalent to the limitations of the study.

Author Response

Dear Reviewer 3,

I’m pleased to resubmit the manuscript of the work entitled, “Novel Digital Measurement Technique to Analyze the Palatine Suture Expansion After Palatine Rapid Maxillary Expansion Technique”

Reviewer 3: English language and style are fine/minor spell check required

Response: In order to adapt to the reviewer's 3 comments, we have send the manuscript to a specialized traductor.

Reviewer 3: The study will make more sense if you can compare the novel technique with a standard technique.

Response: In order to adapt to the reviewer's 3 comments, we have improved the article by comparing the novel digital measurement technique with the previously used lineal measurement technique in order to analyze the repeatability and reproducibility of both techniques.

Reviewer 3: Please be clearer about the null hypothesis. Your hypothesis should answer your primary research question.

Response: In order to adapt to the reviewer's 3 comments, we have we have clarified the null hypothesis

Reviewer 3: Please focus your intro and discussion on the novel technique you are presenting and how is it different from the standard techniques. RME is not the focus here.

Response: In order to adapt to the reviewer's 3 comments, we have added a sentence comparing the novel measurement technique with the previous standard techniques.

Reviewer 3: Please correctly identify the limitations of the study design. Limitations of the technique are not equivalent to the limitations of the study.

Response: In order to adapt to the reviewer's 3 comments, we have identified the limitations of the study.

We take this opportunity to thank the recommendations and suggestions made by the reviewers to improve the document.

Yours sincerely,

Round 2

Reviewer 1 Report

Dear authors,

Thank you for providing all the additional clarifications. The paper is now much more comprehensive to a broader audience. I do not have any additional comments or recommendations. Congratulations on conducting this original study that I had the honor to review.